# Efficient Deep Learning-Based Detection Scheme for MIMO Communication Systems

**DOI:** 10.3390/s25030669

**Published:** 2025-01-23

**Authors:** Roilhi F. Ibarra-Hernández, Francisco R. Castillo-Soria, Carlos A. Gutiérrez, José Alberto Del-Puerto-Flores, Jesus Acosta-Elias, Viktor I. Rodriguez-Abdala, Leonardo Palacios-Luengas

**Affiliations:** 1Faculty of Science, Autonomous University of San Luis Potosí, Av. Chapultepec 1570, Privadas del Pedregal, San Luis Potosí 78295, Mexico; roilhi.ibarra@uaslp.mx (R.F.I.-H.); cagutierrez@ieee.org (C.A.G.); jacosta@uaslp.mx (J.A.-E.); 2Facultad de Ingeniería, Universidad Panamericana, Álvaro del Portillo 49, Zapopan 45010, Mexico; 3Unidad Académica de Ingeniería Eléctrica, Universidad Autónoma de Zacatecas, Carr. Zac-Gdl, Km 6, Zacatecas 98610, Mexico; abdala@uaz.edu.mx; 4Department of Electrical Engineering, Autonomous Metropolitan University (UAM) Iztapalapa, Av. San Rafael Atlixco 186, Iztapalapa, Mexico City 09340, Mexico; lpl@xanum.uam.mx

**Keywords:** MIMO systems, deep learning, detection, labeling, ML criterion, detection complexity, BER performance

## Abstract

Multiple input-multiple output (MIMO) is a key enabling technology for the next generation of wireless communication systems. However, one of the main challenges in the implementation of MIMO system is the complexity of the detectors when the number of antennas increases. This aspect will be crucial in the implementation of future massive MIMO systems. A flexible design can offer a convenient tradeoff between detection complexity and bit error rate (BER). Deep learning (DL) has emerged as an efficient method for solving optimization problems in different areas. In MIMO communication systems, neural networks can provide efficient and innovative solutions. This paper presents an efficient DL-based signal detection strategy for MIMO communication systems. More specifically, a preprocessing stage is added to label the input signals. The labeling scheme provides more information about the transmitted symbols for better training. Based on this strategy, two novel schemes are proposed and evaluated considering BER performance and detection complexity. The performance of the proposed schemes is compared with the conventional one-hot (OH) scheme and the optimal maximum likelihood (ML) criterion. The results show that the proposed OH per antenna (OHA) and direct symbol encoding (DSE) schemes reach a classification performance F1-score of 0.97. Both schemes present a lower complexity compared with the conventional OH and the ML schemes, used as references. On the other hand, the OHA and DSE schemes have losses of less than 1 dB and 2 dB in BER performance, respectively, compared to the OH scheme. The proposed strategy can be applied to adaptive systems where computational resources are limited.

## 1. Introduction

Wireless communications is a pervasive technology whose design involves many distinct challenges for researchers [1,2], such as channel modeling, hardware imperfections, and better signal detection schemes for reliable data reception [3]. Multiple input-multiple output systems (MIMOs) have received special attention as a key enabler technology for 5G/6G wireless communications due to their BER performance gains and higher achievable data rates [4]. As a fundamental part of a MIMO system, signal detection is a procedure that faces high-complexity challenges as the number of transmitter antennas increases. Several researchers have proposed a variety of solutions for MIMO detection systems when computational resources are limited [5,6,7,8,9,10]. Most of strategies achieve near-optimal detection with close performance to that of the maximum likelihood (ML) approach. However, their complexity increases exponentially as the number of transmitter antennas increases. On the other hand, deep learning (DL) is a research area that has gained interest in the last few years. DL is derived from machine learning but is focused on the development of neural network (NN) architectures to solve a variety of problems in engineering, physics, social and economic fields, among others [11,12,13]. As a result of recent advances in big data, optimization techniques and the stronger capacity in computational and hardware resources, DL is seen as an efficient way to design newer solutions for the optimal detection of signals in MIMO systems [14,15].

### 1.1. Related Work

DL-based signal detection for MIMO systems has recently been investigated in different configurations using various modulation schemes to encode symbols during transmission. When the NN architecture considers multiple models, a single training strategy can be used to unify the models, improving the BER performance of the zero-forcing (ZF) and semi-definite relaxation approaches [16]. Motivated by these findings, the authors in [17] present a study that compares the performance of a fully connected NN architecture and a customized DL scheme that unfolds the stochastic gradient descent (SGD) iterations in a network. The authors in [18] find that vector quantization (VQ) presents a similar performance compared to the MIMO-DL detection. This approach uses NN to reduce the quantization loss as the number of transmit (Tx) antennas increases.

DL has also been investigated to improve other MIMO detection approaches, such as sphere decoding [19,20,21] and polar decoding [22,23]. The ML detector is a well-known optimal method that has been implemented with a convolutional neural network (CNN)-based architecture for vehicular networks in a correlated fading channel scenario [24]. In addition, CNN can be applied for the joint design of channel estimation and signal detection in MIMO systems [25]. The CNN model for a MIMO signal detector can also be used in the Internet of Things (IoT) [26,27]. However, a CNN has higher complexity when the levels of the signal-to-noise (SNR) ratio are up to 10 dB [28].

The authors of [29] present different methodologies for the design of a DL-based MIMO detection. The study compares a fully connected long-short-term memory (LSTM) and a CNN-based architecture. For generating labeled data, each index is encoded using a OH strategy for each index. The labeling approach reported in [30] implements Gray encoding when adjacent symbols are presented to help differentiate the loss function. However, if higher-order modulations are required, a DL demodulation with *M*-QAM signals can be considered [31]. In addition, different ML strategies have been implemented for demodulation in MIMO systems, including reinforcement learning [32], and unsupervised learning [33]. Since the multiple NNs designed are not directly connected, several NN-adapted receiver architectures, such as dual-multilevel detection [34] and the design of a parallel NN signal demodulator with the design of a specific loss function, have been proposed. The revised DL-based MIMO signal detection schemes present promising results in BER compared to optimal ML detection. Furthermore, DL-based algorithms have shown highly efficient, robust, and scalable solutions that provide near-optimal performance with lower computational cost, once trained.

However, to the best of our knowledge, there is an absence in the reported results of an appropriate strategy to improve the system considering the training data generated. In this work, we compare two proposals for labeling input data for training DL networks, with the OH strategy in [29] and the optimal ML criterion used in MIMO communication systems.

### 1.2. Contribution

The main contributions of this paper are as follows:We develop two new DL strategies for signal detection in MIMO communication systems. Both strategies perform close to the optimal ML detection scheme.We show that the proposed strategies have a tradeoff between BER performance and complexity and could be used under different hardware constraints.We show that the activation function at the output layer has an impact on the NN-based detector performance.

The remainder of the paper is organized as follows. Section 2 gives a description of the methods used for system design. Section 3 presents the proposed DL detection scheme for MIMO communication systems. Section 4 provides several simulation results and compares the system performance considering related approaches. Finally, Section 5 summarizes the main conclusions and future directions of this research.

*Notation:* Bold-face lowercase letters and bold-face uppercase letters indicate vectors and matrices, respectively. The notations ${\left(\;\cdot \;\right)}^{T}$, ${\left(\;\cdot \;\right)}^{H}$, and ∥·∥ indicate the transpose, the conjugate transpose, and the norm of a vector or matrix. The Gaussian distribution of a complex variable with mean μ and $\sigma^{2}$ variance is denoted as $CN(\mu ,\sigma^{2})$.

## 2. MIMO System Model

In a MIMO system, the signals are transmitted using multiple antennas at the transmitter and receiver. Let s∈A be an *M*-ary modulated symbol, $N_{t}$ the number of Tx antennas, and $N_{r}$ the number of receiving (Rx) antennas, respectively. Then, a vector of multiple symbols $s\in C^{1\times N_{t}}$ transmitted simultaneously in the $\left(N_{r}\times N_{t}\right)$ MIMO systems can be expressed as(1)$$s={\left[\begin{array}{cccc} s_{1}, & s_{2}, & \cdots & s_{N_{t}} \end{array}\right]}^{T}.$$

Let $r\in C^{N_{r}\times 1}$ be the vector of received symbols arriving at each one of the $N_{r}$ receiver antennas, which can be expressed as(2)$$r=\left[\begin{array}{c} r_{1}\\ r_{2}\\ \vdots \\ r_{N_{r}} \end{array}\right]=\left[\begin{array}{cccc} h_{1,1} & h_{1,2} & \cdots & h_{1,N_{t}}\\ h_{2,1} & h_{2,2} & \cdots & h_{2,N_{t}}\\ \vdots & \vdots & \ddots & \vdots \\ h_{N_{r},1} & h_{N_{r},2} & \cdots & h_{N_{r},N_{t}} \end{array}\right]\left[\begin{array}{c} s_{1}\\ s_{2}\\ \vdots \\ s_{N_{t}} \end{array}\right]+\left[\begin{array}{c} n_{1}\\ n_{2}\\ \vdots \\ n_{N_{r}} \end{array}\right],$$
where each element $h_{i,j}\in C$ represents an impulse response of the channel link from the *i*-th transmission antenna to the *j*-th receiving antenna, so each channel response belongs to a channel matrix defined as $H\in C^{N_{r}\times N_{t}}$. Without loss of generality, the evaluation of the system is based on a quasi-static correlated fading channel with Rayleigh distribution where its elements are assumed to be complex Gaussian random variables with zero mean and unit variance, CN(0,1). An equivalent expression of (Equation 2) can be defined as(3)r=Hs+n,
where $s\in C^{1\times N_{t}}$ and $n\in C^{N_{r}\times 1}$ represent the additive white Gaussian noise (AWGN) samples at each *j*-th receiving antenna. We assume that the noise is identically independent distributed with variance $N_{0}$ denoted $CN(0,N_{0})$.

### 2.1. Optimal ML Detection

The optimal ML detection scheme is one of the most common strategies used to evaluate the performance of BER on MIMO systems. The ML detector is a procedure that aims to find the smallest possible distance between the received symbol and all possible products among the estimated channel and the symbol constellation. The optimal ML detection can be defined as(4)$$\hat{s}=\underset{i}{arg min}∥r-\tilde{H}\hat{s_{i}}∥\forall i=1,\cdots ,M^{N_{t}},$$
where $\tilde{H}$ is the estimated channel matrix at the receiver. $\hat{s_{i}}$ is the set of all possible combinations of symbols s. ML is an optimal algorithm, but its implementation is difficult due to its exponential computational complexity proportional to $M^{N_{t}}$ combinations of modulated signals and CSI [35,36]. Without loss of generality, perfect channel state information (CSI) is assumed.

### 2.2. DL-Based MIMO Detection

Let $\phi_{\theta }$ be an NN, composed of *L* layers with initial state $x=x_{0}$. The NN operates by iterating over the following expression:(5)$$x_{\ell +1}=\sigma \left(W_{\ell }x_{\ell }+b_{\ell }\right)\forall \ell =0,\cdots L,$$
where $x_{0}\in R^{N_{F}\times N_{s}}$ stands for the $N_{s}$ training examples of input data with $N_{F}$ parameters used to predict a variable $y\in R^{N_{c}\times N_{s}}$ with $N_{c}$ classes or labels. In Equation (Equation 5), $x_{r}\in R^{d_{\ell }}$ with $\left(d_{0},d_{L+1}\right)=\left(N_{F},N_{c}\right)$, $W_{\ell }\in R^{d_{\ell }+1\times N_{\ell }}$ represents a matrix whose elements are known as the weight coefficients of neurons in each layer, and $b_{\ell }\in R^{d_{\ell +1}}$ is a bias term.

The function σ:R→R in Equation (Equation 5) is an activation function that operates at the output of each layer, introducing nonlinearities in the prediction model [37]. Table 1 shows the three most common choices for the activation function σ(z). First is the rectifier linear unit (ReLU), which introduces nonlinearities to the network so it will be able to model complex patterns. The ReLU function is commonly used in hidden layers due to simplicity in computation. As seen in Table 1, its outputs are zero for negative entries to learn sparse activations. It means that many neurons are inactive to improve computational efficiency and reduce overfitting [38]. The second function is called the sigmoid. A sigmoid is a real, bounded, and differentiable function that provides an output very close to its boundaries. This function is mostly used for binary classification since it maps data to a smooth curve that lies around the interval [0,1]. Sigmoid functions are commonly used in the NN output layer to obtain an output value [39,40]. Finally, softmax activation is a transformation from raw inputs of the network to a vector of probabilities. It is mostly suited for multiclass classification problems, whereas outputs are probabilities that sum up 1. Softmax is also commonly used in the output layer when the final decision is made by evaluating the maximum value of this vector of probabilities [41].

Let $x_{\ell +1}=\hat{y}$ be the predicted output value of the NN. Let $L(y,\hat{y})$ be a loss function. After performing iterations in Equation (Equation 5) with the desired number of *epochs*, the algorithm quantifies the error given by the distance between *y* and $\hat{y}$. The normalized root mean squared error (NRMSE) is defined as(6)$$L\left(y,\hat{y}\right)=\sqrt{\frac{1}{N}\sum_{i=1}^{N}{\left(y_{i}-{\hat{y}}_{i}\right)}^{2}}.$$

NN optimization is performed by finding the minimal values of the loss function, which are convex and differentiable [42]. The stochastic gradient descent (SGD) algorithm is the most commonly used way to find the parameters that optimize the NN. The minimization of a function $f(x_{k})$ follows the iteration(7)$$x_{k+1}=x_{k}-\alpha \nabla f\left(x_{k}\right),$$
where $\nabla f(x_{k})$ is the gradient of the function in the *k*-th iteration and α is a step size called *learning rate*. The fast convergence of SGD can be achieved by adequate preprocessing of the input data, such as normalization and scaling [43].

## 3. Proposed DL-Based Detection Scheme

Figure 1 illustrates the proposed MIMO signal detection scheme based on DL. Throughout the training stage, three labeling encoding strategies are introduced to facilitate accurate signal detection. In the first stage, the information bits are assigned to the *M*-QAM symbols s. Then, a serial-to-parallel converter sends each symbol to a transmission antenna $a_{k}$, with $k=\{1,2,\cdots ,N_{t}\}$. Each symbol is multiplied by a channel impulse response $h_{i,j}\in H$ arriving at any reception antenna, where the AWGN is added. The symbols received $r_{j}\in r$, with $j=\{1,2,\cdots ,N_{r}\}$ are divided into their real and imaginary parts $R(r_{j})$ and $I(r_{j})$, respectively; therefore, the input training data contain $N_{F}=4$ features.

Algorithm 1 describes the general procedure of the proposed MIMO DL detection scheme. First, we take the real and imaginary parts of each received symbol r as the input data. Then, we select a strategy to generate the matrix y to label each entry row of the data array X. During the training stage, we perform the forward pass calculation of $x_{\ell +1}$ over several epochs. The values of the weights and biases array $W_{\ell }$ and $b_{\ell }$ are optimized when performing the SGD method from the calculation of its gradients $\nabla W_{\ell },\nabla b_{\ell }$, respectively. For the monitoring and evaluation of the training loop, we calculate the LMSE loss function $L(y,\hat{y})$. Once the weights and biases arrays have been optimized (the training phase loop has ended), we perform *forward pass* again to estimate the received symbol $\hat{r}$. Now, it is possible to obtain the BER using the designed model.
**Algorithm 1** Proposed DL-based detector.**Intput:** 
$s,N_{t},N_{r},L,N_{e},W_{\ell },b_{\ell },\sigma \left(\;\cdot \;\right),\alpha$
**Output:** 
$\hat{r},L\left(y,\hat{y}\right)$
 Symbol reception:
r=Hs+n
 Generate input data
$X:\left\{R\left(r_{j}\right),I\left(r_{j}\right)\right\}\;\forall j=1,\cdots ,N_{r}$
 Generate y labels vector by selecting any label encoding method
 $x_{0}=X\;$                          ▷ Start of training phase
 **for** $k=1,\cdots ,N_{e}$ 
**do**
    $x_{\ell +1}=\sigma \left(W_{\ell }x_{\ell }+b_{\ell }\right)\;\forall \ell =0,\cdots ,L$
    Calculate gradients: $\nabla W_{\ell },\nabla b_{\ell }$
    $W_{\ell }:=W_{\ell }-\alpha \;\cdot \;\nabla W_{\ell }$                      ▷ SGD optimization    $b_{\ell }:=b_{\ell }-\alpha \;\cdot \;\nabla b_{\ell }$
    $\hat{y}=x_{L}$
    Calculate RMSE loss: $L(y,\hat{y})$
 **end for**                           ▷ End of training phase
 $\hat{r}=\sigma \left(W_{\ell }x_{\ell }+b_{\ell }\right)\;\forall \ell =0,\cdots ,L$             ▷ Received symbol estimation


From the revised DL-based frameworks in literature, it can be seen that the information and data processing of the received symbols r cannot be extended to extract more learning parameters. Therefore, in this framework, our main approach is to explore different ways of labeling the predicted class vector y. This is actually how to assign values to the classifier that identify the M-QAM symbol and antenna labels to feed the DL-based detector. The procedure is described in Figure 1 as label encoding, which is executed only during the training phase. We analyze the following three labeling strategies.

One-hot (OH) encoding of the $M^{N_{t}}$ combinations in transmission [29].Direct symbol encoding (DSE) $(\log_{2}\left(M\right)\times N_{t}$ different encoding labels).One-hot encoding per antenna (OHA) for the possible *M* symbols ($M\times N_{t}$ labels).

The first stage of the label encoding algorithm (Figure 1) is the mapping rule for the symbols to bit labels (opposed to modulation). For example, for a 4-QAM modulation, we assign a bit to each real and complex part of the symbols according to the sign value (positive or negative). Table 2 shows the binary representation of the labels for the 4-QAM constellation, which is the reference for the three analyzed labeling encoding cases.

After obtaining the input data $x_{0}=\left\{R\left(r_{j}\right),I\left(r_{j}\right)\right\}$ and selecting any of the label encoding choices for y, we feed a neural network which consists of *L* fully-connected hidden layers, each activated with a ReLU activation function. However, we select different activation functions for the output layer according to the label encoding method to improve the detection of symbols. Throughout the training stage, the evaluation of the model is also carried out. The proposed MIMO signal detection consists of recovering the transmitted M-QAM symbols; therefore, it is considered a supervised learning classification task. The following metrics are commonly used for measuring the performance in DL-based classification tasks:(8)$$\mathrm{Precision}=\frac{TP}{TP+FP},$$(9)$$\mathrm{Recall}=\frac{TP}{TP+FN},$$(10)$$F1=\frac{2\times \mathrm{Precision}\times \mathrm{Recall}}{\mathrm{Precision}+\mathrm{Recall}},$$
where TP is the number of positive values or correctly classified samples, FP is the number of false positive values, and FN is the number of false negatives. Note that in this case, multiclass classification, we need to evaluate an average value for each performance metric. The F1-score is calculated as the harmonic mean between precision and recall. It offers a balanced perspective of performance that adapts to different scenarios [44].

### 3.1. Direct Symbol Encoding (DSE)

In the DSE labeling strategy, we take the bit labels defined in Table 2, where each two-bit label for each Tx antenna is concatenated. An example considering a 4-QAM modulation is presented in Table 3. First, we indicate the original symbol index and then concatenate the corresponding label as referenced in Table 2. The number of different class labels is the number of possible combinations $Nc=\log_{2}\left(M\right)\times N_{t}$. The activation function that provides better performance in this case is a sigmoid since the entries in Table 3 are binary but contain multiclass values, that is, more than one value of the binary label contains a “1” bit in all cases. This label encoding is useful when the available hardware and processing capacity are limited.

### 3.2. One Hot Encoding (OH)

The OH labeling strategy consists of encoding each index of one combination of symbols as an OH vector, where each entry provides a vector of zero entries except for the position of the index, where the bit label is encoded as “1”. The output is the vector y with dimensions $Nc=2^{N_{t}}$, corresponding to the number of classes. Table 4 shows an example of this OH encoding per index for a MIMO system that contains $N_{t}=2$ antennas and 4-QAM modulation. This is the highest-dimensional case for the three proposed methods. The activation function that is well suited for this strategy is softmax since the last decision will be taken evaluating the maximum value of the vector of probabilities obtained at the output of the softmax activation. This label encoding is recommended when there are no hardware and processing capacity constraints in the system.

### 3.3. One Hot Encoding per Antenna (OHA)

The OH labeling represents the approach of the highest dimension because it requires an exponential number of possible values. However, when the OH encoded input is divided into the $N_{t}$ antennas, this dimensionality is reduced to $N_{c}=M\times N_{t}$ different class values.

Table 5 shows an example of the OHA method. In this case, $N_{t}=2$ transmission antennas and 4-QAM modulation are considered. The first *M* bits correspond to the OH vector for the first antenna, where the symbol index is indicated as the position of the binary value of “1” in each vector. For example, when both antennas $a_{1}$ and $a_{2}$ transmit the symbol $s_{1}$, the corresponding bit label is [10001000]. We propose this labeling encoding procedure when the hardware requirements are higher than in the case of symbol encoding but lower than OH encoding. Although OH fits better with softmax, this labeling sigmoid activation is a better option since the encoded vectors contain $N_{t}$ entries encoded as “1”, having a pattern better recognized for this activation.

## 4. Results and Discussion

This section describes the setup of the experiments carried out in this work. We present and discuss the results. Table 6 synthesizes the experiments. The first setup is a 2×2 MIMO system, which consists of two transmitter and two receiver antennas. The NN architecture for signal detection consists of L=1 hidden layer with 100 hidden units. The second setup consists of a 4×4 MIMO system, with $N_{t}=4$ and $N_{r}=4$ respectively. Detection is achieved using an NN with L=2 hidden layers, each containing 1000 neurons. The layers are fully connected for both cases.

During the training phase, for all designed DL detectors, the learning rate selected is α=0.01, the Xavier method is used to initialize the weights, and the NRMSE (defined in Equation (Equation 6)) is the loss function. The experiments are conducted in MATLAB©, Julia, and Octave frameworks on a workstation with 128 RAM GB, Intel Xeon Gold 5220R 24 cores @2.20 GHz, and Ubuntu 22.04 LTS operative system. The first architecture for MIMO 2×2 is trained with $N_{e}=2000$ epochs and for the 4×4 two-layer NN, the number of epochs is increased to $N_{e}$ = 50,000. The symbols used for the training dataset are modulated with 4-QAM. The channel samples follow a Rayleigh distribution, where perfect CSI at the receiver is assumed and the received samples are equalized. For simulations, an SNR level of 3 dB is selected.

Figure 2 shows the NMSE loss curves obtained during the training phase for the MIMO configurations 2×2 and 4×4. In both cases, the training and test subsets present a close decay among epochs. Since there is no huge gap between the train and test datasets’ NMSE curves, overfitting is not present among all models. It can be observed that OH labeling curves have the best BER performance since NMSE decays faster than the other approaches. The DSE strategy model shows the worst performance. For the 4×4 MIMO configuration, the NMSE cannot be reduced as in the 2×2 configuration between epochs. In the last training epochs, the NMSE curve presents oscillations rather than decay. The OHA strategy for labeling in the MIMO DL detector is an intermediate option for both cases.

Figure 3 shows a comparison of the BER performance for the 2×2 MIMO DL schemes. The OH encoding scheme presents the closest performance to the optimal ML detector at every level of SNR. The OHA shows performance with 0.5 dB of distance compared to the optimal ML detector. Finally, the DSE shows a distance of almost 2 dB compared with the reference scheme for a $\mathrm{BER}=10^{-4}$.

Figure 4 shows the BER performance of the systems for the 4×4 MIMO configuration. Since the number of antennas increases the dimensionality, the number of neurons and layers also increases, compared to the MIMO 2×2 configuration. The behavior of the BER performance curves is similar to the 2×2 configuration. The OH encoding presents the closest distance to the optimal ML detector, with 0.5 dB approximately. The next closest BER curve is the OHA, which now presents a gap of almost 1 dB compared to the ML detector. Finally, the DSE scheme presents the furthest curve to ML performance, with a difference of almost 2 dB at $\mathrm{BER}=10^{-4}$.

Furthermore, the performance of the MIMO detectors is evaluated considering the classification metrics previously defined in Section 2. Table 7 shows the results of this evaluation. The best performance is achieved when the OH labeling approach is selected in both MIMO configurations, with 0.97 and 0.95 F1-scores, respectively. This method also has the highest precision and recall values. Nevertheless, the results shown in Table 7 indicate that the performance metrics are directly related to the number of classes $N_{c}$, which is also supported by the fact that the worst results are obtained when using the DSE strategy, where $N_{c}$ is the lowest among all labeling methods. However, for DSE, the classification metrics outcomes are around 0.9, showing a labeling strategy with good performance. The OHA strategy produces classification performance values above symbol encoding and the OH strategy.

### Complexity Analysis

The ML detector defined by Equation (Equation 4) computes the Euclidean distance between the received vector r and the products of the CSI matrix H with all the combinations possible $M^{N_{t}}$ received of the modulated symbols s. This procedure involves $M^{N_{t}}$ products, $N_{t}\;\cdot \;N_{r}$ subtractions, and norm evaluation that results in a complexity of $O(N_{t}\;\cdot \;N_{r}\;\cdot \;M^{N_{t}})$. In the case of DL detectors, we add the multiplications produced at each layer to obtain their complexity. The operations produced by activation functions can be neglected, as the calculations of the outputs are not affected by the input size. Table 8 lists the resulting complexities for all DL-based detection schemes. For this evaluation, we consider only the forward propagation since the backward propagation has a similar complexity. Therefore, the total complexity for each system is approximately twice the values listed in Table 8 if the training phase is considered. Figure 5 shows the resulting complexity in flops for the signal-labeling and detection strategies designed in this paper.

The results show that complexity is directly affected by the number of hidden units in each layer $d_{\ell +1}$. This value is multiplied by twice the number of Rx antennas $N_{r}$ due to the input size of $x_{0}$, and it will not affect the final result. The number of Tx antennas $N_{t}$ and the modulation order *M* have different variations in complexity regarding the selected labeling strategy.

The OH labeling scheme provides the highest complexity and the best BER performance, F1-score, precision, and recall metrics. On the other hand, the lowest complexity is achieved when using a DSE strategy to label signals, reducing the last factor to $\log_{2}\left(M\right)$ and obtaining the lowest BER performance but not having the worst classification score. The OHA provides an alternative to reduce the complexity from exponential $M^{N_{t}}$ to a product $M\;\cdot \;N_{t}$ with promising results in BER performance.

## 5. Conclusions

In this paper, different DL-based approaches for signal detection in MIMO communications systems have been presented. An efficient alternative to processing the input data by structuring different labeling strategies to the Tx signals has been introduced. We evaluated the impact of these labeling strategies on the system performance. The of BER curves showed that the proposed schemes perform close to the optimal ML detection criterion with a simple NN architecture assuming perfect CSI. The proposed DL-based detectors presented distances in BER curves of almost 0.5 dB when applying the conventional OH encoding as a labeling method. Nonetheless, the proposed DL signal detection schemes showed lower complexity compared to the optimal ML criterion. In our proposal, the complexity was reduced to $M\;\cdot \;N^{t}$ for the case of OHA, and $\log_{2}M\;\cdot \;N_{t}$ when applying the DSE strategy. The classification performance metrics in terms of F1-score above 0.9 also indicated that the designed DL schemes are promising for the detection of 4-QAM modulated signals. Among the DL-based evaluated schemes, the OH encoder showed the best BER performance. However, it requires more resources as the dimensionality and computational complexity increase. The proposed OHA and the DSE schemes showed a BER performance very close to that of the OH scheme with the advantage of reduced complexity when the number of Tx antennas increases. These approaches are convenient for systems with hardware or signal processing constraints, mainly in massive MIMO designs. Future research improvements include testing higher-order *M*-QAM modulation schemes and hyperparameter tuning to optimize system resources. Furthermore, different channel estimation strategies will be considered.

## Figures and Tables

**Figure 1 sensors-25-00669-f001:**
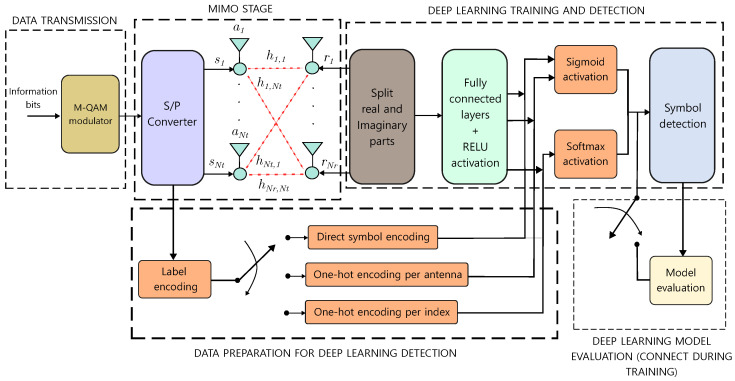
Proposed DL-based detection scheme for MIMO communication systems.

**Figure 2 sensors-25-00669-f002:**
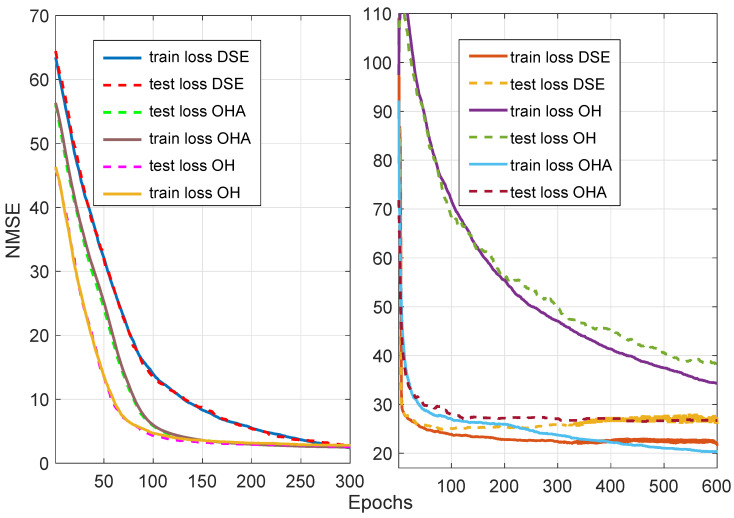
NMSE loss curves vs the number of epochs during the training phase for the MIMO 2×2 (**left**) and 4×4 (**right**) DL-based detection configurations. The dashed lines indicate the NMSE values for the test or validation subsets, while the solid lines are the NMSE values for the training subset.

**Figure 3 sensors-25-00669-f003:**
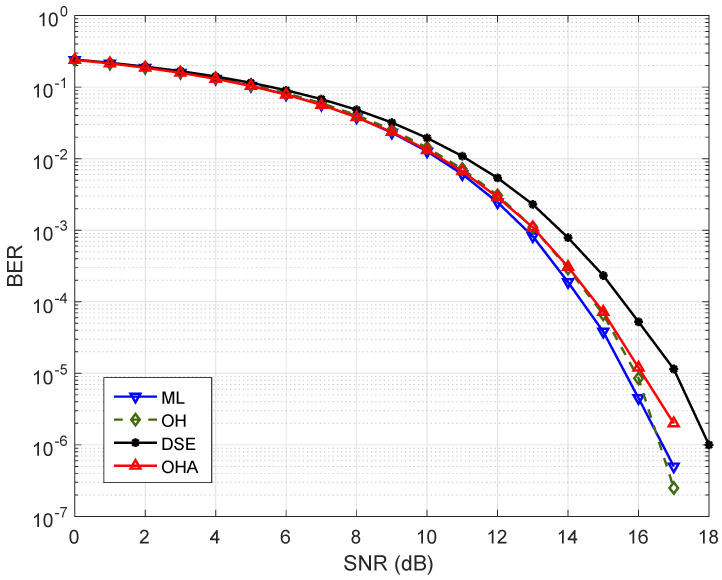
BER performance of the detectors for the analyzed MIMO detection schemes with $N_{t}=2$ and $N_{r}=2$ antennas. The optimal ML detector and DL–based detectors with the OH, the OHA, and the DSE of symbol labeling strategies are applied.

**Figure 4 sensors-25-00669-f004:**
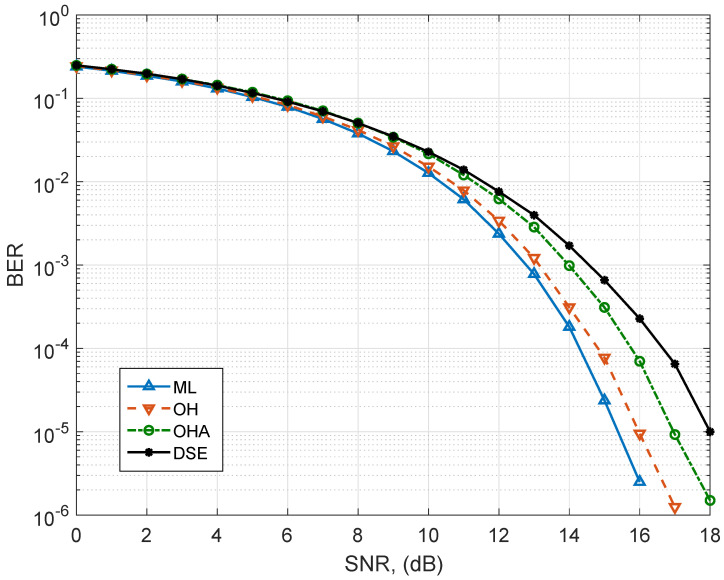
BER performance of the detectors for the analyzed MIMO detection schemes with $N_{t}=4$ and $N_{r}=4$ antennas. The optimal ML detector and DL–based detectors with the OH, the OHA, and the DSE labeling strategies are applied.

**Figure 5 sensors-25-00669-f005:**
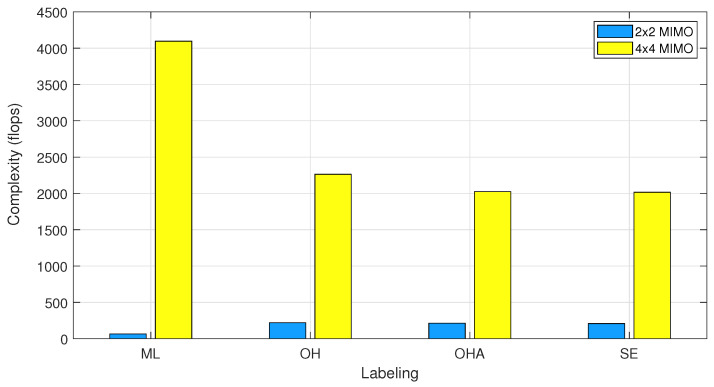
Complexity in flops for the developed DL-detectors. The labeling strategies acronyms are ML: maximum likelihood, OH: one-hot encoding, OHA: one-hot encoding per antenna, SE: symbol encoding.

**Table 1 sensors-25-00669-t001:** Activation functions commonly used for NN architectures.

Function Name	Computation	Typical Use
ReLU	max(0,z)	Hidden layers
Sigmoid	$\frac{1}{1+e^{-z}}$	Output layer
Softmax	$\frac{e^{z_{k}}}{\sum_{k}e^{z_{k}}}$	Output layer

**Table 2 sensors-25-00669-t002:** Reference binary encoding of labels for 4-QAM modulation symbols.

Symbol Label	Complex Value	Bit Label Mapping
$s_{1}$	−1+1j	1 0
$s_{2}$	−1−1j	1 1
$s_{3}$	1+1j	0 0
$s_{4}$	1−1j	0 1

**Table 3 sensors-25-00669-t003:** DSE labeling of symbols using as reference Table 1. MIMO system with $N_{t}=2$ and 4-QAM modulation.

Transmitted Symbol		Bit Label Mapping
$a_{1}$	$a_{2}$	
$s_{1}$	$s_{1}$	1 0 1 0
$s_{1}$	$s_{2}$	1 0 1 1
$s_{1}$	$s_{3}$	1 0 0 0
$s_{1}$	$s_{4}$	1 0 0 1
$s_{2}$	$s_{1}$	1 1 1 0
$s_{2}$	$s_{2}$	1 1 1 1
$s_{2}$	$s_{3}$	1 1 0 0
$s_{2}$	$s_{4}$	1 1 0 1
$s_{3}$	$s_{1}$	0 0 1 0
$s_{3}$	$s_{2}$	0 0 1 1
$s_{3}$	$s_{3}$	0 0 0 0
$s_{3}$	$s_{4}$	0 0 0 1
$s_{4}$	$s_{1}$	0 1 1 0
$s_{4}$	$s_{2}$	0 1 1 1
$s_{4}$	$s_{3}$	0 1 0 0
$s_{4}$	$s_{4}$	0 1 0 1

**Table 4 sensors-25-00669-t004:** OH encoding for each index combination of symbols. Example case for $N_{t}=2$ antennas and 4-QAM modulation.

Symbol Transmitted		Bit Label Mapping
$a_{1}$	$a_{2}$	
$s_{1}$	$s_{1}$	1 0 0 0 0 0 0 0 0 0 0 0 0 0 0 0
$s_{1}$	$s_{2}$	0 1 0 0 0 0 0 0 0 0 0 0 0 0 0 0
$s_{1}$	$s_{3}$	0 0 1 0 0 0 0 0 0 0 0 0 0 0 0 0
$s_{1}$	$s_{4}$	0 0 0 1 0 0 0 0 0 0 0 0 0 0 0 0
$s_{2}$	$s_{1}$	0 0 0 0 1 0 0 0 0 0 0 0 0 0 0 0
$s_{2}$	$s_{2}$	0 0 0 0 0 1 0 0 0 0 0 0 0 0 0 0
$s_{2}$	$s_{3}$	0 0 0 0 0 0 1 0 0 0 0 0 0 0 0 0
$s_{2}$	$s_{4}$	0 0 0 0 0 0 0 1 0 0 0 0 0 0 0 0
$s_{3}$	$s_{1}$	0 0 0 0 0 0 0 0 1 0 0 0 0 0 0 0
$s_{3}$	$s_{2}$	0 0 0 0 0 0 0 0 0 1 0 0 0 0 0 0
$s_{3}$	$s_{3}$	0 0 0 0 0 0 0 0 0 0 1 0 0 0 0 0
$s_{3}$	$s_{4}$	0 0 0 0 0 0 0 0 0 0 0 1 0 0 0 0
$s_{4}$	$s_{1}$	0 0 0 0 0 0 0 0 0 0 0 0 1 0 0 0
$s_{4}$	$s_{2}$	0 0 0 0 0 0 0 0 0 0 0 0 0 1 0 0
$s_{4}$	$s_{3}$	0 0 0 0 0 0 0 0 0 0 0 0 0 0 1 0
$s_{4}$	$s_{4}$	0 0 0 0 0 0 0 0 0 0 0 0 0 0 0 1

**Table 5 sensors-25-00669-t005:** OHA scheme at each Tx antenna. $N_{t}=2$ antennas and 4-QAM modulation.

Symbol Transmitted		Bit Label Mapping
$a_{1}$	$a_{2}$	
$s_{1}$	$s_{1}$	1 0 0 0 1 0 0 0
$s_{1}$	$s_{2}$	1 0 0 0 0 1 0 0
$s_{1}$	$s_{3}$	1 0 0 0 0 0 1 0
$s_{1}$	$s_{4}$	1 0 0 0 0 0 0 1
$s_{2}$	$s_{1}$	0 1 0 0 1 0 0 0
$s_{2}$	$s_{2}$	0 1 0 0 0 1 0 0
$s_{2}$	$s_{3}$	0 1 0 0 0 0 1 0
$s_{2}$	$s_{4}$	0 1 0 0 0 0 0 1
$s_{3}$	$s_{1}$	0 0 1 0 1 0 0 0
$s_{3}$	$s_{2}$	0 0 1 0 0 1 0 0
$s_{3}$	$s_{3}$	0 0 1 0 0 0 1 0
$s_{3}$	$s_{4}$	0 0 1 0 0 0 0 1
$s_{4}$	$s_{1}$	0 0 0 1 1 0 0 0
$s_{4}$	$s_{2}$	0 0 0 1 0 1 0 0
$s_{4}$	$s_{3}$	0 0 0 1 0 0 1 0
$s_{4}$	$s_{4}$	0 0 0 1 0 0 0 1

**Table 6 sensors-25-00669-t006:** MIMO setup for the conducted experiments.

MIMO Configuration	Hidden Layers	Number of Neurons
2×2: $N_{t}=2$, $N_{r}=2$	1	100
4×4: $N_{t}=4$, $N_{r}=4$	2	1000

**Table 7 sensors-25-00669-t007:** Classification performance for the analyzed MIMO detection schemes.

MIMO Configuration	Labeling Approach	Precision	Recall	F1-Score
$N_{t}=2$ $N_{r}=2$	OH	0.97	0.98	0.97
	OHA	0.93	0.93	0.93
	DSE	0.96	0.98	0.96
$N_{t}=4$ $N_{r}=4$	OH	0.96	0.94	0.95
	OHA	0.89	0.89	0.91
	DSE	0.91	0.92	0.93

**Table 8 sensors-25-00669-t008:** Complexity of the analyzed DL-based MIMO signal detection schemes.

MIMO Configuration	Labeling Approach	Complexity
$N_{t}=2$ $N_{r}=2$	OH	$O(\left(2\left(N_{r}+d_{\ell +1}\right)+M^{N_{t}}\right))$
	OHA	$O(\left(2\left(N_{r}+d_{\ell +1}\right)+M\;\cdot \;N_{t}\right))$
	DSE	$O(\left(2\left(N_{r}+d_{\ell +1}\right)+\log_{2}\left(M\right)\;\cdot \;N_{t}\right))$
$N_{t}=4$ $N_{r}=4$	OH	$O(d_{\ell +1}\left(2Nr+M^{N_{t}}\right))$
	OHA	$O(d_{\ell +1}\left(2N_{r}+M\;\cdot \;N_{t}\right))$
	DSE	$O(d_{\ell +1}\left(2N_{r}+\log_{2}\left(M\right)\;\cdot \;N_{t}\right))$

## Data Availability

New data were not generated in this research. However, the source code for reproducing the results is publicly available at https://github.com/roilhi/mimo-dl-detector (Accessed date: 18 December 2024).

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
