# Peer review of "Efficient Deep Learning-Based Detection Scheme for MIMO Communication Systems"

_sensors, 2025, doi:10.3390/s25030669_

Round 1

Reviewer 1 Report

Comments and Suggestions for Authors

The paper addresses a MIMO communication system, proposing DL-based detection strategies to enhance signal detection performance. I have one minor comment. Fig.1 should be improved cause I can not see the deep learning process in this figure.

Author Response

Dear Editor and anonymous Reviewers,

We appreciate your time and valuable feedback on our manuscript. All your comments are highly constructive and help us enhance the quality of the paper. In the lines below, we give our answer to the comments received. Please note that updates have been highlighted in yellow in the manuscript, and each response has been typed in blue to facilitate their identification.

Francisco R. Castillo Soria, Ph. D.

January 10, 2025

*Please see the response to Reviewer 1 in the attached file.

Reviewer 2 Report

Comments and Suggestions for Authors

1.     Some of the references used are more than 5 years old. I would suggest replacing them with recent works.

2.     The abstract provides a good overview of the research, but it could be more interesting if you consider the following elements for clarity and completeness:

·       A brief explanation of MIMO technology and its significance in wireless communication could help set the stage for the research.

·       Elaborate on why the proposed deep learning strategy is necessary. What specific challenges in traditional MIMO designs does it address?

·       A more detailed description of the preprocessing stage and the two proposed schemes would clarify how they differ from existing methods.

·       While the abstract mentions an F1-score, including specific numerical results for BER performance and complexity comparisons could strengthen the impact.

·       Discussing the broader implications of the findings for future MIMO systems or potential applications in real-world scenarios would enhance the abstract's relevance.

3.     The background and literature works could be enhanced further as this domain is reach with plenty of manuscripts online.

4.     Equations need to be cited.

5.     In the simulation section, description of figures could be enhanced.

6.     The result need to be compared with a couple of previous works from the references you have mentioned so as to validate its novelty.

Author Response

Dear Editor and anonymous Reviewers,

We appreciate your time and valuable feedback on our manuscript. All your comments are highly constructive and help us enhance the quality of the paper. In the lines below, we give our answer to the comments received. Please note that updates have been highlighted in yellow in the manuscript, and each response has been typed in blue to facilitate their identification.

Francisco R. Castillo Soria, Ph. D.

January 10, 2025

* Please see the response to Reviewer 2 in the attached file.

Reviewer 3 Report

Comments and Suggestions for Authors

This paper is related to MIMO, that is a key technology for the next generation of wireless communication systems, especially to MIMO detection algorithms. Many algorithms have been proposed for MIMO systems, this paper proposes detection algorithms based on Deep learning.

The paper seems to be rather interesting to specialists in this field, but I suggest the following comments and remarks to improve it.

Please give more explanations to terms "conventional one-hot scheme" and "F1-score of 0.97", this is not entirely clear.

In Abstract section additional keywords can be added: ML criterion, detection complexity, BER performance.

In Introduction section several detection approaches are mentioned, but it is not entirely clear, why the proposed DL strategy needed. There is effective Sphere decoding algorithm and other near-optimal approaches. All K-best algorithm should be mentioned, because it has reduced complexity. 

In MIMO system model (equations (1) - (3)) please indicate if this is a complex case or a real one, e.g. information symbols "s" have real or complex values, channel matrix is real or complex, etc. If the noise is complex, please check its variance. 

In equation (4) variable "i" is not defined.

For Section 2.2. Conventional detection can also mean MMSE approach.

I suppose for Table 1 functions could be described in more detail, or references to the literature could be given.

In section 3 "Proposed DL-based detection scheme" please outline the main differences and advantages of the proposed approach compared to known DL detection algorithms.

In Section 4. "Results and discussion" please specify what do You mean - 2 x 2 (4 x 4) number of antennas or layers, do You suppose Beamforming or not? In modern systems much more antennas are used (Massive MIMO is mentioned in the article).

For Figures 2 - 4 can You add if it is possible other detection results to compare (e.g. MMSE, K-Best, other close DL approach). The same - for Figure 5 (Complexity analysis).

In Conclusion please specify what known and what alternative approaches were considered, and what is the BER and Complexity improvement compared to known approached. Future research improvements can be directed to more number of antennas, non-ideal CSI, maybe additional complexity reduction can be achieved - please take this inti account in the Conclusion section. I suppose slight correction of English could be made.

Author Response

Dear Editor and anonymous Reviewers,

We appreciate your time and valuable feedback on our manuscript. All your comments are highly constructive and help us enhance the quality of the paper. In the lines below, we give our answer to the comments received. Please note that updates have been highlighted in yellow in the manuscript, and each response has been typed in blue to facilitate their identification.

Francisco R. Castillo Soria, Ph. D.

January 10, 2025

* Please see the response to Reviewer 3 in the attached file.

Round 2

Reviewer 3 Report

Comments and Suggestions for Authors

The authors improved the manuscript, I have no any comments.
Maybe some minor checking/correction of the text is needed.

Author Response

Dear Editor,

The authors of this paper would like to thank you for your support and the comments received. Thanks to your comments and those of the reviewers, this paper has been significantly improved.

In the new version, this manuscript has been carefully revised. Several typographical errors and 
spelling mistakes have been corrected. Furthermore, the wording of some paragraphs has been improved. These changes have been pointed out in the text.

Sincerely,

Francisco R. Castillo Soria
Corresponding author.